# Mass drug administration coverage and its determinants for the elimination of onchocerciasis in Ulanga District, Tanzania

**Ambakisye Kuyokwa Mhiche**[1,2,3]*, **Dinah Gasarasi**[2], **George Kabona**[3], **Ally Hussein**[4], **Upendo John Mwingira**[1], **Ahmed Mohamed Abade**[5]

1 Research Triangle Institute International, Dodoma, Tanzania, 2 Muhimbili University of Health and Allied Sciences, Dar es Salaam, Tanzania, 3 Neglected Tropical Diseases Control Program, Ministry of Health, Dodoma, Tanzania, 4 Tanzania Field Epidemiology and Laboratory Training Program, Ministry of Health, Dar Es Salaam, Tanzania, 5 African Field Epidemiology Network (AFENET), Dar Es Salaam, Tanzania

* ambakisyekuyokwa@ymail.com

## Abstract

### Background

Onchocerciasis remains an important public health problem targeted for elimination in Tanzania. Ulanga District was hyperendemic for onchocerciasis before the intervention, underscoring the need for particularly high coverage in MDA efforts. The district has been implementing MDA through Community Directed Treatment with Ivermectin (CDTI) strategy since 1998. However, there is preliminary evidence of persistent onchocerciasis transmission, which likely sustains the observed high prevalence of Onchocerciasis in both human and vector species. This could be linked to poor treatment coverage. This study was conducted to assess treatment coverage and explore factors that determine drug uptake during MDA program.

### Methods

A cross-sectional study was conducted in Ulanga District, Morogoro, Tanzania, from April to June 2019. Using multistage cluster sampling, 502 participants were randomly selected and interviewed through a structured questionnaire. Modified Poisson regression analysis was used to identify independent factors affecting MDA uptake.

### Results

The overall response rate was 96% with majority (67%) of respondents being females, and the mean age of the study participants was 37.8 years with a standard deviation (SD) of ± 15 years. The study found that MDA coverage varied across villages: Mawasiliano (68%, CI: 59.3 – 75.6), Uponera (83%, CI; 76.6 – 89.6), Isongo (84.8%, CI: 77.3 – 90.1) and Togo (79%, CI: 70.1 – 85.8). While Uponera and Isongo achieved the WHO-recommended 80% coverage for transmission interruption, Mawasiliano and Togo fell below this threshold. Age significantly influenced drug uptake, with younger individuals (15–24 years) having a lower uptake rate [APR = 2.8, p = 0.008], followed by the 25–34

**Data availability statement:** All relevant data are available within the manuscript and its Supporting Information files.

**Funding:** The author(s) received no specific funding for this work.

**Competing interests:** The authors have declared that no competing interests exist.

age group [APR = 2.3, p = 0.04]. Occupation also played a role, as small and medium enterprise (SME) workers [APR = 3.2, p = 0.05] and students [APR = 2.9, p = 0.05] were less likely to participate. Residence duration in the village was a strong predictor of MDA uptake. Individuals living in the village for more than a year were significantly more likely to participate [APR = 2.3, p = 0.00]. Additionally, a lack of awareness about MDA benefits negatively impacted participation, as those uncertain about its benefits were less likely to take the drug [APR = 2.5, p = 0.03]. Similarly, individuals unaware of the correct MDA distribution schedule had lower uptake [APR = 2.5, p = 0.03]. However, those who took Ivermectin for prevention purposes were significantly more likely to participate [APR = 13.4, p = 0.001].

## Conclusion

MDA coverage below the WHO optimally recommended coverage has been demonstrated in the villages studied. This implies low drug uptake, delayed interruption of transmission and Onchocerciasis elimination. The findings highlight the need for targeted interventions to improve MDA coverage by focusing on younger individuals, certain occupational groups, and new residents. Strengthening community engagement, improving health communication, and intensifying biannual MDA efforts are recommended to accelerate Onchocerciasis elimination in the district. It also underscores the need for adopting other effective public health interventions such as community mobilization towards slash and clear of potential breeding sites for Onchocerciasis vector.

## Author summary

Onchocerciasis remains a public health concern in Tanzania, despite over two decades of mass drug administration (MDA) with ivermectin. This study examines the MDA coverage in Ulanga District and identifies factors influencing drug uptake. Using a cross-sectional survey of 502 participants, we found that treatment coverage varied across villages, with some failing to reach the WHO-recommended 80% threshold for transmission interruption. Key barriers to MDA participation included younger age, recent migration, certain occupations (e.g., students, small business workers), and lack of awareness about MDA benefits and scheduling. Fear of ivermectin's side effects and misconceptions further reduced uptake. However, individuals who believed in ivermectin's preventive benefits were significantly more likely to participate. Our findings highlight the need for improved health communication, targeted engagement strategies for young and migrant populations, and consideration of alternative treatments to expand coverage to excluded groups such as pregnant women and children under five. Addressing these challenges through enhanced community mobilization and tailored interventions is crucial to achieving onchocerciasis elimination. These insights provide valuable guidance for optimizing MDA programs in Tanzania and other endemic regions striving to meet elimination targets

## Introduction

Onchocerciasis (river blindness) is a parasitic disease transmitted through the bites of black flies from the Simulium species, caused by the parasite *Onchocerca volvulus* [1]. The flies

breed in fast-flowing waters of streams and rivers, most notably in Africa, including Tanzania [2]. Onchocerciasis control relies on the Community Directed Treatment with Ivermectin (CDTI) strategy, which involves annual mass drug administration (MDA) of Ivermectin. Improvement in the ivermectin treatment coverage during MDA have been linked to successful transmission interruption and prevalence reduction in several Onchocerciasis endemic areas [3–5]. Eliminating onchocerciasis involves attaining 100% geographical coverage in all areas with ongoing transmission, achieving and sustaining the WHO-recommended MDA treatment coverage of 80% of the total population, and demonstrating the interruption of transmission among vector species [6].

Although treatment coverage is an important factor for onchocerciasis elimination process, it only highlights the prevailing situation at community level and does not necessarily guarantee good adherence to drug uptake by individuals in the community. Studies have shown that, individuals who don't take ivermectin each year provide a sources of reinfection for their communities [7]. Therefore, for successful control and elimination of Onchocerciasis, all residents in the high-risk population even those who are apparently in good health must take the drugs during the MDA campaign [6,8–10].

In Tanzania, Onchocerciasis remains an important public health problem and it is among the five most prevalent preventive chemotherapy Neglected Tropical Diseases (PC-NTDs) targeted for elimination in the country [11,12]. Substantial progress in the control of the diseases has been made and a goal to eliminate Onchocerciasis by 2025 has been set [13]. However, to achieve this, community adherence to drug uptake during MDA campaign for high treatment coverage is paramount.

Ulanga District within Mahenge Onchocerciasis foci in Tanzania was known for its high endemicity since 1960s and has been implementing MDA through CDTI strategy since 1998 [14]. A baseline assessment for onchocerciasis conducted in 1997 indicated nodule prevalence ranging 45% to 95% in Ulanga districts [14]. Various studies revealed increased prevalence of Epilepsy (nodding Syndrome) in Ulanga that implies presence of relatively high level of filaria worms in the rea, suggesting limited progress made in the control of Onchocerciasis [15–18]. Despite two decades of MDA in the area transmission is still ongoing raising uncertainty on its possibility to attain elimination goal by 2025 [19]. According to 2009 and 2017 survey data, Ulanga had an overall maximum prevalence of 22%, Onchocerciasis nodule prevalence was 2.3%, while the positivity rate for the OV16 rapid test was 76.5%, and the seroprevalence among children aged 6–9 years was 20.7% [20]. This high seroprevalence in young children, considering that onchocerciasis is a chronic infection that can persist for over 10 years, suggests that there was still significant O. volvulus transmission at the time of the study. Existence of co-endemic conditions such as Epilepsy implies presence of relatively high level of filaria worms in the area, suggests limited progress made towards elimination [20]. On the other hand, literature shows that the black fly in Ulanga District carries infective parasites, with an infection rate of 0.57%, which is nearly twelve times higher than the transmission interruption threshold of 0.05% as set by the WHO [21]. Therefore, this study was designed to assess the coverage and determinants for drug uptake for Ivermectin MDA in Ulanga district in order to gather appropriate evidence to inform all stakeholders involved in the endeavor to fight Onchocerciasis in the country.

## Methods

### Ethical consideration

Ethical clearance for this study was obtained from the Muhimbili University of Health and Allied Sciences Research and Publications Ethical Committee (Ref. No. DA.287/298/01A).

Research permits were secured from the Regional Administrative Secretary for Morogoro region and the District Executive Director for Ulanga District Council. Written informed consent was obtained from all willing respondents prior to data collection. For participants aged 15–18 years, written informed consent was also obtained from their parents or guardians. Semi-structured and in-depth interviews were conducted in private settings to ensure confidentiality. Respondents who chose not to participate after providing informed consent were excluded from the study without any impact on their future participation in mass drug administration (MDA) campaigns. Confidentiality was maintained throughout data collection; no names or addresses were recorded in the mobile data collection application. Only the IT Programmer and Principal Investigator had password-protected access to the data on the server.

## Study area

The study was conducted in Morogoro region within Mahenge Onchocerciasis focus at Ulanga district council. The area is in southeastern Tanzania; it is a mountainous area with fast-flowing perennial rivers which are Luli, Mbalu, Lukande, Mzelezi, Ruaha and Msingizi Rivers that serves as the potential breeding sites for black flies. The 2012 census recorded 265,203 residents in Ulanga district. Administratively the district has been divided into 3 divisions and 31 wards whereby each ward has about three and above villages. In 2017, the OV16 prevalence in the district was 76% in rural areas and 50% in suburban areas. Evidence of active transmission among children aged 6–9 years has been recorded to be 20.7% [20]. This study was condicted in four villages nalemy Isongo, Uponera, Mawasiliano, and Togo. These villages were classified as rural, where onchocerciasis prevalence tends to be higher.

## Study design

This was a cross-sectional community-based study using a multistage cluster sampling method involving both secondary data review and survey carried out in Ulanga district, Morogoro region from April-June 2019.

## Study population

All individuals living in the study area were considered the source population for this study. The study focused on community members aged 15 years and above living in households of the sampled villages. Individuals who had been approached by a community drug distributor (CDD) during the last year before the study was conducted and participated in the MDA were eligible if were able to explain themselves clearly. The study excluded Individuals who were exempted during MDA campaign because of ineligibility such as being below 5 years old, pregnant women, and those with serious health problems.

## Sample size estimate

The multistage cluster sampling method was employed, with the number of clusters and design effects for cluster sampling determined using Steve Bennett's methodology. The sample size was calculated utilizing the Kish-Leslie formula for a single proportion cross-sectional study. Sample size calculation considered the assumption that proportion of drug uptake during MDA in Mahenge would be 78% of the total population as determined previously by 2017 study. With a 5% level of precision, standard critical value of 1.96 with 95% confidence interval, and non-response rate of 10%.

Design effects and number of clusters were determined using Steve Bennett's formula as shown below [22].

$$D = 1 + (b-1) \text{ X roh}$$

where; D = Design effect, b = expected number of households covered in each sub-village (20) and roh = Rate of homogeneity among clusters (0.1).

Then, D = 1+ (20 - 1) x 0.1 hence D = 2.9.

Number of clusters (Hamlet/Sub-village) was calculated using Steve Bennett's formula [22] given below

$$c = p(1-p)d \, / \, \text{s2b}$$

where, C = Number of clusters (hamlet/sub-villages) that will be included in the study, p = Proportion drug uptake in Ulanga (78%), D = Design effect (2.9), s = Level of precision (5%) and b = expected number of households covered in each sub-village (35).

Then, C = 0. 78 x (1-0.78) x 2.9 and 0.052 x 35 hence C = 5.7 ~ 6.

The sample size was determined using the Kish-Leslie Formula for a single proportion cross-sectional study, as outlined below.

$$N = \text{gZ2P}(1-p) \, / \, d^2$$

where: N = Sample size, Z = standard critical value, P = Proportion drug uptake in Ulanga and d = The acceptable margin of error to be 5%.

Then; N = [2.9 x 1.962 x 0.78 (1-0.78)] ÷ 0.0652, N = [2.9 x 1.962 x 0.78x 0.22] ÷0.0652 hence N = 453.

After adjusting to the non-response rate of 10%, 502 was obtained as the minimum sample size.

In addition to the community survey, secondary data was obtained from community Drug Distributors (CDD) registers to explore a three-year trend in MDA uptake among community members during MDA campaign.

## Data collection procedures

The study involved primary data collection households randomly selected from 4 stage cluster sampled villages. First stage: Ulanga District council was selected as one of the councils in Mahenge Onchocerciasis foci with high active transmission of Onchocerciasis despite ongoing MDA intervention. Second stage: a list of onchocerciasis endemic wards in Mahenge division was obtained from DMO office, two wards were randomly selected. Third stage: a list of the villages from the selected wards were obtained and four villages were randomly selected. From which, 6 hamlets were selected according to Steve Bennett's formula. Fourth stages: households from each selected hamlet were randomly selected with starting point at the centre of the hamlet; each eligible household member was prospectively followed until the anticipated sample size of 502 was attained.

Secondary data collection was done in 2 health facilities within the selected villages. A tool to assess the MDA uptake was developed to collect information among those who consented to participate. Apart from questionnaire, a checklist for data abstraction from community drug distributors register (community register) was also developed. An Open Data Kit (ODK Collect) mobile application freely available in google play (https://play.google.com/store/apps/details?id=org.odk.collect.android&hl=en&gl=US) was developed and used to collect data. To ensure the validity of our study, we used standardized questionnaire, that were pre-tested to assess the adequacy of the interviewer-based questionnaire and the accuracy of mastering data

entry using a mobile phone devise. For the survey, content validity was established by designing the questionnaire based on recognized frameworks and expert input, while construct validity was ensured by aligning the survey teams with the study objectives. Reliability was strengthened through internal consistency checks and pilot testing.

For the qualitative guide, validity was maintained by developing questions informed by literature and expert consultation to accurately capture key themes. Reliability was enhanced through interviewer training and data source triangulation.

The necessary permissions to conduct the study were sought from respective authorities (region, district, ward and village governments). Data were collected using mobile devices through Open Data Kit (ODK Collect) application. Questionnaire were uploaded into mobile phone, corrected and checked for quality within the respective mobile devises daily by the Information Technology (IT) personnel from The National NTDs Control Program. Review of all Community Drug Distributors register in the selected villages was done to determine individuals in the community who participated in the past MDA campaigns using a checklist. Interviewer administered semi structured questionnaire was used to collect information on factors that determine drug uptake in the community. The questionnaire had three (3) major parts; part A had 7 questions that explored the socio-demographic characteristics of the participants. Part B had 13 questions that explored the community perceived disease and drug effects that influence drug uptake especially treatment effects, perceived knowledge, belief and perception about Ivermectin drugs. Part C had 3 questions that looked at the program expert support, awareness creation and drug supply system for ivermectin distribution particularly community sensitization both formal and informal in facilitating individual decision on taking of ivermectin drugs during MDA campaign. Data completeness and accuracy was checked on daily basis and any ambiguities were immediately addressed the following day during the study period April to June 2019.

## Data management and analysis

Data were transferred to the NTDCP server from the mobile phone (Smartphone) after crosschecking their completeness and accuracy. From the server data were extracted as excel files that could be easily read by Epi Info and STATA analysis software. Data was cleaned and analyzed using both Epi Info 7.2.2.6 by CDC Atlanta, Georgia (US) and Stata 15 by College Station, Tx:StataCorpLLC. Confidence Intervals around proportions were calculated using Fleiss Quadratic approximation formula. We reported at 95% CI level using the Wald method which uses the normal approximation to the binomial distribution and ideal for a larger sample size. Following descriptive analysis, generalized linear model with modified Poisson regression was used to estimate prevalence ratio. Prevalence ratio refers to the prevalence of taking ivermectin during MDA and that it is an interchangeable term with "relative risk". The variables were further subjected to Poisson regression model with negative binomial link robust. The variables with $p \leq 0.05$ were regarded as determinants of drug uptake. To control confounding effect all factors with $p < 0.25$ in bivariate analysis were subjected to modified Poisson regression. MDA drug Uptake, Socio demographic, drug consumer behavior, CDD behavior and IVM treatment factors were analyzed and summarized in Tables 1–3.

## Results

A total of 502 participants were recruited during the study period with a response rate of 96%. The majority (67%) were females, and the mean age of the study participants was 37.8 years with a standard deviation (SD) of ±15 years. The mode of the participants was 25–34 (25.5%). MDA coverage for the studied villages was as follows; Mawasiliano (68%, CI: 59.3

– 75.6), Uponera (83%, CI; 76.6 – 89.6), Isongo (84.8%, CI: 77.3 – 90.1) and Togo (79%, CI: 70.1 – 85.8). While Uponera and Isongo Villages had reached the WHO recommended optimal coverage (80%), the remaining villages (Mawasiliano and Togo) was below the optimal coverage recommended by WHO (80%) for successful transmission interruption. Drug uptake coverage observed in Isongo, Togo and Uponera was significantly different for the three-year period. While Mawasiliano had consistently a low uptake across all three years with no significant changes. In Isongo village, the percentage of individuals taking MDA drugs fluctuated over the years. In 2016, 67.1% participated, but the uptake dropped significantly to 45.8% in 2017 before rising again to 63.3% in 2018. The prevalence ratio (PR) in 2017 was 2.41 [PR = 2.41 (95% CI: 2.2-2.3), p = 0.001], indicating a significant difference from 2016. In 2018, the PR was still statistically significant [PR = 1.18 (95% CI: 1.1- 1.3), p = 0.002], meaning that uptake remained higher than in 2017 but was still not as strong as in 2016.

On the other hand, Uponera Village Uponera Village showed a steady increase in drug uptake, starting at 58.1% in 2016 and rising to 61.5% in 2018. The PR in 2017 was 0.99 [PR = 0.99 (95% CI: 0.89-1.12), p = 0.93], showing no significant difference from 2016. However, in 2018, the PR was 0.87 [PR = 0.87 (95% CI: 0.78-0.96), p = 0.02] indicating statistically significant improvement in drug uptake.

Togo exhibited a stable drug uptake trend with minor variations. The participation rate increased from 65.9% in 2016 to 70.7% in 2017, with a PR of 0.8 [PR = 0.8 (95% CI: 0.7-0.9), p = 0.003]. However, in 2018, the uptake slightly declined to 68.0%, with a PR of 0.9 [PR = 0.9 (95% CI: 0.8-1.0), p = 0.13] suggesting that the difference was not statistically significant (Table 1).

## Socio demographic factors associated with MDA drug uptake

A stepwise modified Poisson regression model was developed that included Age, Education level, Marital Status, Occupation and Duration of stay in the villages to determine their

**Table 1. Comparison of Drug uptake along three-year period in selected villages in Ulanga District.**

| Village | MDA Drug Uptake | | | |
|---|---|---|---|---|
| | Yes (%) | No (%) | PR (95%CI) | p-value |
| **Isongo** | | | | |
| 2016 | 1570 (67.1) | 812 (32.9) | 1 | 1 |
| 2017 | 1691 (45.8) | 701 (54.2) | 2.41 (2.2-2.3) | 0.001* |
| 2018 | 2018 (63.3) | 1671 (36.7) | 1.18 (1.1- 1.3) | 0.002* |
| **Togo** | | | | |
| 2016 | 1570 (65.9) | 812 (34.1) | 1 | 1 |
| 2017 | 1691 (70.7) | 701 (29.3) | 0.8 (0.7-0.9) | 0.003* |
| 2018 | 1671 (68.0) | 787 (32.0) | 0.9 (0.8-1. 0) | 0.13 |
| **Mawasiliano** | | | | |
| 2016 | 555 (22.6) | 1899 (77.4) | 1 | 1 |
| 2017 | 578 (22.8) | 1952 (77.2) | 0.99 (0.86-1.13) | 0.85 |
| 2018 | 599 (23.0) | 2001 (77) | 0.98 (0.86-1.11) | 0.72 |
| **Uponera** | | | | |
| 2016 | 1399 (58.1) | 1006 (41.9) | 1 | 1 |
| 2017 | 1438 (58.3) | 1029 (41.7) | 0.99 (0.89-1.12) | 0.93 |
| 2018 | 1559 (61.5) | 976 (39.5) | 0.87 (0.78-0.96) | 0.02* |

*PR= Prevalence ratio,

*CI= Confidence Interval.

association with drug uptake. Age and duration of stay in the village among participants were found to be independent predictors of drug uptake. Younger individuals, particularly those aged 15–24 and 25–34, had lower uptake rates, as indicated by their Adjusted Prevalence Ratio [APR = 2.8 (95% CI: 1.3-6.2), p < 0.05)] and [APR = 2.3 (95% CI: 1.04-4.9), p < 0.05)] respectively. This suggests that these age groups are less likely to participate in MDA programs compared to those aged 45–54, who serve as the reference group. However, for individuals aged 35–44 and 55–93, the results were not statistically significant, indicating that age-related trends in uptake may primarily affect younger individuals

Occupation plays a significant role in determining MDA uptake. Small and Medium Entrepreneurs (SMEs) workers and students had substantially lower uptake rates compared to extension workers, who served as the reference category. SMEs had an APR of 3.2 [APR = 3.2 (95% CI:2. 0.99-10.6), p < 0.05)] while students had an APR of 2.9 [APR = 2.9 (95% CI: 0.9-9.8), p < 0.05). Although the effects for students were marginally significant, they were still notable. These results suggest that occupational status influences MDA participation, with those in informal or student roles being less likely to participate.

The length of time spent in the village was an indicator of MDA drug uptake. Individuals who had lived in the village for more than one year were significantly more likely to participate in MDA programs [APR = 2.3 (95% CI: 1.5-3.4), p < 0.05)] compared to those who had stayed for less than a year. These findings highlight the importance of long-term residence in fostering participation, possibly due to greater integration into community health initiatives and awareness programs.

Marital status appears to have some influence on MDA uptake, with unmarried individuals showing a slightly higher crude prevalence ratio (CPR = 1.2, p = 0.009) compared to married individuals. However, after adjusting for other variables, this effect could not sustain statistical significance (APR = 1.2, p = 0.39). This suggests that while marital status may initially appear to impact uptake, other factors, such as occupation or age, may be more critical in explaining the differences observed. Similarly, Education level did not appear to be a strong determinant of MDA drug uptake. The results show that individuals with no education, secondary education, or college education had similar uptake rates to those with primary education. None of the adjusted prevalence ratios (APR) for education categories were statistically significant, indicating that formal education does not strongly predict MDA participation. This finding suggests that other factors, such as awareness campaigns or community influence, may play a larger role than education in determining uptake (Table 2).

## Perceived disease and drug effects and their influence on drug uptake

Several factors such as pre-MDA sensitization, MDA distribution cycle, perceived benefit of IVM in symptoms control, understanding MDA distribution interval and perceived reason for drug uptake among others) were investigated as to understand their association with drug uptake among communities in Ulanga.

One notable finding was the significant negative effect of uncertainty regarding MDA benefits. Individuals who responded "Don't Know" when asked about MDA benefits were significantly less likely to participate, with an APR of 4.7 [APR = 4.7 (95% CI: 1.03-21.1), p = 0.05]. This suggests that lack of awareness or confidence in the benefits of MDA is a major barrier to uptake, indicating that lack of knowledge about IVM's benefits significantly impacts drug uptake.

Fear of side effects negatively affected drug uptake [CPR = 1.8 (95% CI: 1.4-2.6), p = 0.00]. However, after adjustment, the association was not significant [APR = 1.2 (95% CI: 0.9-1.3), p = 0.08], indicating that while fear influences decision-making, other factors like awareness

**Table 2. Socio demographic characteristic associated with MDA drug uptake in multi variate analysis, in Ulanga district, June 2019.**

| Variable | MDA Drug Uptake | | | | | |
|---|---|---|---|---|---|---|
| | Yes (%) | No (%) | CPR (95%CI) | p-value | APR (95%CI) | p-value |
| **Age:** | | | | | | |
| 15-24 | 72 (64.3) | 40 (35.7) | 3.9 (1.9-8.3) | 0.00* | 2.8 (1.3-6.2) | 0.008* |
| 25-34 | 94 (76.4) | 29 (23.6) | 2.6 (1.2-5.6) | 0.02* | 2.3 (1.04-4.9) | 0.04* |
| 35-44 | 80 (82.50) | 17 (17.5) | 1.9 (0.8-4.4) | 0.12 | 1.9 (0.8-4.3) | 0.13 |
| 55 - 93 | 64 (87.7) | 9 (12.3) | 1.4 (0.5- 3.5) | 0.52 | 1.3 (0.5-3.4) | 0.53 |
| 45-54 | 70 (90.9) | 7 (9.1) | 1 | 1 | 1 | 1 |
| **Education Level:** | | | | | | |
| None | 16 (69.6)) | 7 (33.4) | 1.4 (0.7- 3.9) | 0.25 | 1.7 (0.6-4.6) | 0.29 |
| Secondary | 59 (67.1) | 29 (32.9) | 1.1 (0.7-1.7) | 0.72 | 1.1 (.6-2.2) | 0.68 |
| College | 8 (66.7) | 4 (33.3) | 1.7 (0.7-8.2) | 0.16 | 2.1 (0.4-10.4) | 0.37 |
| Primary | 297 (82.7) | 62 (27.3) | 1 | 1 | 1 | 1 |
| **Marital Status** | | | | | | |
| Married/Cohabiting | 230 (83) | 47 (17) | 1 | 1 | 1 | 1 |
| Not Married | 150 (73.2) | 55 (26.8) | 1.2 (1.2-2.8) | 0.009* | 1.2 (0.8-1.7) | 0.39 |
| **Occupation** | | | | | | |
| Peasant | 341 (81.2) | 79 (18.8) | 1.9 (0.4-8.2) | 0.42 | 1.8 (0.6-5.4) | 0.30 |
| SMEs | 14 (58.3) | 10 (41.7) | 5.7 (1.1-30.6) | 0.04* | 3.2 (0.99-10.6) | 0.05 |
| Student | 9 (45) | 11 (55) | 9.8 (1.8-54.3) | 0.009* | 2.9 (0.9-9.8) | 0.09 |
| Extension Worker | 16 (88.9) | 2 (11.1) | 1 | 1 | 1 | 1 |
| **Duration of stay in the village** | | | | | | |
| Below one year | 10 (37) | 17 (63) | 1 | 1 | 1 | 1 |
| One year and above | 370 (81.3) | 85 (18.7) | 7.4 (3.3-16.7) | 0.0001* | 2.3 (1.5-3.4) | 0.00* |

*CPR = Cruse Prevalence Ratio,

*APR = Adjusted Prevalence Ratio,

*CI = Confidence Interval.

and community support may counterbalance this concern. Similarly, individuals who knew others who had previously taken ivermectin showed no significant difference in uptake (APR = 1.1, p = 0.15), suggesting that personal experience alone does not strongly predict uptake.

On the other hand, those who were unaware of the correct MDA distribution interval were significantly less likely to take the drug [APR = 2.5 (95% CI:.1-6.0), p = 0.03]. This indicates a strong link between knowledge of MDA timing and participation and emphasizing the need for better communication about MDA schedules.

A notable finding was that individuals who were concerned about alcohol restrictions were significantly more likely to take the drug [APR = 12 (95% CI: 2.4-60.9), p = 0.03]. This suggests that despite known concerns about the effects of alcohol when taken with medications, there appears to be limited concern in the community, which may serve as a significant behavioral driver.

Likewise, those who took ivermectin for prevention were much more likely to participate [APR = 13.4 (95% CI:.12.9-60.9), p = 0.001], reinforcing that perceived health benefits drive MDA uptake. In contrast, those who experienced a single side effect from previous drug use still had high participation (98.3%). This suggests that prior experience, even if negative, may not deter future participation but rather reinforce the importance of taking preventive measures.

Individuals who know the symptoms of onchocerciasis had a higher crude prevalence ratio [CPR = 5.0, (95% CI: 3.7-6.8), p < 0.001], meaning they are significantly more likely to participate in MDA drug uptake. However, after adjusting for other factors, the association become not statistically significant [APR = 0.9, (95% CI: 0.8-1.1), p = 0.38], suggesting that knowledge of symptoms alone does not drive participation. Attending pre-MDA advocacy and sensitization sessions strongly influences drug uptake. Those who did not attend had a significantly higher CPR of 4.7, [CPR = 4.7, (95% CI: 2.8-7.1), p < 0.001C], indicating a lower likelihood of participation. However, after adjusting for other factors, the association becomes non-significant [APR = 1.02, (95% CI: 0.8-1.3), p = 0.85]. This indicates that attending sensitization programs alone might not be the key factor influencing drug uptake, implying other factors influence participation beyond advocacy alone.

Health education sources significantly influenced MDA drug uptake. Those who received information through TV and radio had a [CPR = 5, (95% CI: 1.2-21.2), p = 0.03], while those who never heard about it had an even higher [CPR = 11, (95% CI: 2.8-43.5), p = 0.00]. After adjusting for other factors, not hearing about MDA remained marginally significant [APR = 1.6, (95% CI: 0.9-2.5), p = 0.07]. This suggests that in rural areas targeting community sensitization through radio and TV may not be an ideal approach (Table 3).

## Discussion

Despite 20 years of Mass Drug Administration (MDA) implementation in Ulanga district, drug uptake remains suboptimal. This study sought to assess reported treatment coverage and identify programmatic and sociodemographic factors that influence MDA participation in order to better understand the reasons for ongoing transmission that suggest persistent infection.

Although individuals aged 15–24 years showed lower participation in MDA, there were no significant changes in uptake among those aged 35–44 and ≥ 55 compared to the 45–54 age group. The lower compliance among the 15–24 age group could be linked to their limited experience with the disfiguring effect of the disease on the human body. However, as the age increases, there is a noticeable increase in MDA participation, suggesting an increase in compliance. This trend contrasts with findings from other studies, which reported no significant differences in uptake across age groups [23,24]. Educating children about MDA drug distribution and its health benefits has been associated with better community comprehension of the effects of ivermectin delivered through MDA. Evidence suggests that educating children is crucial, as it encourages them to adhere to treatment and share this knowledge with their families, contributing to the sustained success of MDA efforts [25]. The low participation of younger children in the CDTI program in Ulanga district aligns with studies that report a high prevalence of epilepsy, including nodding syndrome, in the area [15,16]. Higher MDA uptake among younger groups could have potentially reduced nodding syndrome incidence due to ivermectin effects.

Lower uptake among participants who had lived in the area for less than one year may be due to insufficient knowledge of onchocerciasis and a low perception of risk, as well as limited awareness of the MDA campaign. This finding aligns with other studies that have shown low drug uptake among residents who had lived in a Community-Directed Treatment with Ivermectin (CDTI) area for less than five years [26]. Ulanga's natural resources, such as fertile land and mineral deposits including spinel, gold, ruby, and graphite, attract migrants. These high rates of migration could explain the presence of many non-native residents, whose engagement in MDA programs may be limited. Failure to address these dynamics during MDA campaigns could hamper the effectiveness of the intervention and result in suboptimal coverage.

Table 3. Factors associated with MDA drug uptake in multi variate analysis, in Ulanga district.

| Variable | MDA Drug Uptake | | CPR (95%CI) | p-value | APR (95%CI) | p-value |
|---|---|---|---|---|---|---|
| | Yes | No | | | | |
| **Understanding Onchocerciasis Symptoms** | | | | | | |
| Knows symptoms | 357 (86.6) | 55 (13.4) | 5.0 (3.7-6.8) | 0.00* | 0.9 (0.8-1.1) | 0.38 |
| Don't know the symptoms | 23 (76.7) | 47 (23.3) | 1 | 1 | 1 | 1 |
| **Pre MDA advocacy and sensitization attended** | | | | | | |
| No | 208 (92.9) | 16 (7.1) | 4.7 (2.8-7.1) | 0.00* | 1.02 (0.8-1.3) | 0.85 |
| Yes | 172 (66.3) | 86 (33.3) | 1 | 1 | 1 | 1 |
| **Onchocerciasis Targeted health education** | | | | | | |
| Hygiene and Ocho | 118 (98/3) | 2 (1.7) | 0.2 (0.05-1.13) | 0.07 | 0.4 (0.2-1.2) | 0.12 |
| Other but not Ocho | 78 (57.1) | 6 (42.9) | 6 (2.2-16.0) | 0.01* | 0.6 (0.2-1.7) | 0.33 |
| Don't remember | 176 (66.7) | 88 (33.3) | 4.7 (2.1-10.3) | 0.00* | 0.6 (0.2-1.8 | 0.36 |
| Onchocerciasis only | 78 (92.9) | 6 (7.1) | 1 | 1 | 1 | 1 |
| **Perceived impact of advocacy attended in MDA drug uptake.** | | | | | | |
| Didn't Influenced | 195 (97.5) | 5 (2.5) | 1 | 1 | 1 | 1 |
| Influenced next MDA | 185 (65.6) | 97 (34.4) | 13.8 (5.7-33.2) | 0.00* | 2.1 (0.7-6.4) | 0.17 |
| **Source Pre-uptake Health Education** | | | | | | |
| FLHW | 184 (94.4) | 11 (5.6) | 1.4 (0.3-6.2) | 0.65 | 1.5 (0.8-2.6) | 0.18 |
| TV and Radio | 52 (80) | 13 (20) | 5 (1.2- 21.2) | 0.03* | 1.6 (0.9-2.5) | 0.07 |
| Never heard | 96 (55.8) | 76 (44.2) | 11 (2.8-43.5) | 0.00* | 1.6 (0.9-2.6) | 0.05 |
| Reference CDD | 48 (96) | 2 (4) | 1 | 1 | 1 | 1 |
| **Benefit of IVM in symptoms control** | | | | | | |
| Stop Itching | 325 (97.3) | 9 (2.7) | 1 | 1 | 1 | 1 |
| Don't Stop Itching | 37 (97.4) | 1 (2.6) | 0.97 (0.12-7.5) | 0.98 | 2.6 (0.3-20.4) | 0.38 |
| Don't know | 18 (16.4) | 92 (83.6) | 31 (16.2.2-59.5) | 0.00* | 4.7 (1.03-21.1) | 0.05* |
| **Perceived fear of IVM side effects** | | | | | | |
| Influenced drug uptake | 83 (67.5) | 40 (32.5) | 1.8 (1.4-2.6) | 0.00* | 1.2 (0.9-1.3) | 0.08 |
| Not influenced drug uptake | 297 (82.7) | 40 (17.3) | 1 | 1 | 1 | 1 |
| **Experience of previously treated patient** | | | | | | |
| Influenced drug uptake | 244 (75.8) | 78 (24.2) | 1.6 (1.1-2.5) | 0.02* | 1.1 (0.9-1.3) | 0.15 |
| Not influenced drug uptake | 136 (85) | 24 (15) | 1 | 1 | 1 | 1 |
| **Understanding MDA distribution Interval** | | | | | | |
| Once a year | 311 (96.6) | 11 (3.4) | 1 | 1 | 1 | 1 |
| After every 6 months | 62 (89.9) | 7 (10.1) | 2.9 (1.2- 7.4) | 0.02* | 1.3 (0.3-5.5) | 0.73 |
| Don't Know | 7 (7.7) | 82 (92.3) | 27 (15.1-48.5) | 0.00* | 2.5 (1.1-6.0) | 0.03* |
| **History of previous effects following drug uptake** | | | | | | |
| At least two | 267 (88.1) | 36 (18.9) | 1 | 1 | 1 | 1 |
| One Effect | 113 (63.1) | 66 (36.9) | 3.1 (2.1- 4.5) | 0.00* | 0.9 (0.8-1.1) | 0.56 |
| **Perceived reason for drug uptake:** | | | | | | |
| Fear of alcohol restriction | 81 (92) | 7 (8) | 4.8 (1.5-14.6) | 0.007* | 12 (2.4-60.9) | 0.003* |
| Prevention Effects | 5 (5.3) | 90 (94.7) | 56.7 (23.7-135.4) | 0.00* | 13.4 (2.9-60.9) | 0.001* |
| Side effects | 294 (98.3) | 5 (1.7) | 1 | 1 | 1 | 1 |

*CPR = Cruse Prevalence Ratio,

*APR = Adjusted Prevalence Ratio,

*CI = Confidence Interval.

The study found that MDA coverage in Ulanga district remains below the 80% threshold recommended by the World Health Organization (WHO) for effective transmission interruption [27]. Other studies have similarly reported low treatment coverage in long-established MDA programs, identifying this as a key barrier to achieving transmission interruption [15]. Addressing these challenges and improving treatment coverage is crucial to the success of MDA programs.

Interestingly, drug uptake was higher among participants who believed in the preventive effects of ivermectin on onchocerciasis (p < 0.001). This increased confidence in the environment could be due to the reduced number of people with visible symptoms of onchocerciasis in the community, reinforcing the perceived effectiveness of the drug. Similar results were reported in Cameroon and Burkina Faso, where three-quarters of participants complied with CDTI, believing in ivermectin's health benefits [8,28]. However, the study also found that fear of ivermectin's side effects, driven by community misconceptions such as fears of male impotence, birth control effects, and swelling was associated with low drug uptake. These myths and misconceptions contribute to the reluctance of some community members to participate in preventive services [24]. Knowledge of MDA distribution cycles was identified as an important factor influencing drug uptake. Participants who were aware of the cycles were more likely to participate in multiple rounds compared to those who lacked such knowledge. This finding contrasts with earlier research, which reported no significant association between knowledge of MDA rounds and drug uptake. The lack of awareness campaigns and inadequate community sensitization prior to drug distribution may have contributed to this gap. Intensive health education and community engagement could significantly improve participation in future MDA rounds [29]. It is possible that intrinsic factors, such as population mobility and livelihood activities, were overlooked during previous MDA campaigns, contributing to the consistently low coverage reported in Ulanga. These factors must be better addressed in future MDA efforts to achieve the recommended coverage and accelerate progress towards onchocerciasis control and elimination.

The exclusion of pregnant women, children under five years, and other vulnerable individuals in the community from MDA programs increases the likelihood of not reaching optimal coverage, posing significant challenges to achieving the ambitious goal of onchocerciasis elimination by 2025. These untreated groups serve as potential sources of continued transmission. This issue could be addressed with new drugs or drug combinations that have a greater effect on *Onchocerca volvulus* and demonstrate higher safety levels in human populations [30]. Exploring the possibility of treating children as young as four years with alternative microfilaricidal drugs, like moxidectin, to complement ivermectin which can only be administered to those five years and older could help increase coverage among younger children, enhance microfilaria clearance, and accelerate progress toward elimination [31].

Additionally, given that pregnant women are often excluded from ivermectin treatment for several years and may serve as infection reservoirs, the development of safe alternative treatments targeting pregnant women and postnatal mothers would ensure they receive treatment and help prevent further transmission of the disease [17,18].

## Study limitation

The main limitation of this study was the fact that investigations were conducted four to five months after the last MDA distribution paving a way for the recall bias. Nevertheless, it is believed that the results concerning the treatment coverage and compliance to drug uptake are reliable, because the methodological approach included review of all the records available in the community register of the study area and cross checking with the records at the

 

district level. Also, the uniqueness of ivermectin tablets themselves in terms of both physical appearance (small and white) and their distribution strategy makes the treatment cycle easily remembered.

## Conclusion

Despite over two decades of MDA implementation in Ulanga district, the uptake of ivermectin remains suboptimal, threatening efforts to interrupt the transmission of onchocerciasis. From 2018 the district embarked into biannual MDA for Ivermectin to accelerate elimination. This study has identified key sociodemographic and programmatic factors influencing low compliance, particularly among younger age groups and newly settled individuals. While younger people (aged 15–24) showed lower participation rates, older adults exhibited higher compliance, potentially due to increased awareness of the disease's effects. Additionally, individuals living in the district for less than a year showed reduced participation, likely due to limited knowledge of the disease and the MDA campaign.

Low coverage remains a significant barrier to achieving the WHO-recommended threshold of 80% necessary for transmission interruption. Factors such as misconceptions about the side effects of ivermectin, lack of awareness about MDA cycles, and the exclusion of certain vulnerable groups (pregnant women and children under five) further exacerbate these challenges. However, belief in the preventive benefits of ivermectin was positively associated with drug uptake, highlighting the potential impact of effective health education and community sensitization.

## Recommendations

Enhance Education and Sensitization Efforts: Develop targeted health education programs aimed at younger individuals to increase their awareness of onchocerciasis risks and the benefits of MDA participation. But also Implement school-based interventions to educate children and adolescents, encouraging them to participate in MDA and to advocate for treatment within their families.

Engage Migrant and Non-Native Populations: Adapt MDA campaigns to effectively reach recent migrants and non-native residents, possibly through tailored messaging and inclusion strategies that consider language and cultural differences. As well as collaborate with local leaders and organizations to identify and engage these populations.

Address Misconceptions and Fears: Conduct community dialogues and workshops to dispel myths and misconceptions about ivermectin's side effects. Use testimonials and success stories from community members who have benefited from MDA to build trust and confidence in the program. Leverage the community directed approached such as GESI (Gender Equity and Social Inclusions (GESI) through iDARE (Identify gaps and Create Solutions for community for GESI related barrier) methodology to ensure every individual in the community participate in the CDTI programs.

Include Vulnerable Groups in MDA Programs: Explore the use of safe and effective alternative treatments for pregnant women and children under five, such as moxidectin or other suitable drugs. Advocate for policy changes that allow for the inclusion of these groups in MDA efforts, ensuring comprehensive coverage.

Increase Awareness of MDA Distribution Cycles: Improve communication about the timing and frequency of MDA rounds through community meetings, local media, and engagement with community health workers. Encourage consistent participation by emphasizing the importance of completing multiple treatment cycles for effective disease control.

Adapt MDA Strategies to Local Context: Consider intrinsic factors such as population mobility and livelihood activities when planning MDA campaigns. Schedule MDA activities

 

at times that are convenient for the majority of the population, including those involved in agriculture or mining.

Implement New Treatments and Drug Combinations: Support research and integration of new drugs or drug combinations with higher efficacy and safety profiles, such as moxidectin, to improve treatment outcomes. Pilot alternative treatment regimens that could reduce the treatment duration and accelerate progress toward elimination.

Strengthen Community Engagement: Foster partnerships with local leaders, healthcare providers, and community-based organizations to enhance the reach and effectiveness of MDA programs. Utilize community-directed treatment approaches to empower residents and promote ownership of onchocerciasis elimination efforts.

By implementing these recommendations, MDA programs in Ulanga district can improve drug uptake, especially with the ongoing biannual treatment thus achieve optimal coverage and make significant strides toward the elimination of onchocerciasis.

## Supporting information

**S1 Data. Cleaned dataset that was used to generate the study findings.**
(XLSX)

**S2 Data. Coverage table that was used to develop figures.**
(XLSX)

## Acknowledgments

We are immensely grateful to the Neglected Tropical Diseases Control Program for availing space in the data base and equipment for electronic data collection.

## Author contributions

**Conceptualization:** Ambakisye Kuyokwa Mhiche.

**Data curation:** Ambakisye Kuyokwa Mhiche.

**Formal analysis:** Ambakisye Kuyokwa Mhiche.

**Investigation:** Ambakisye Kuyokwa Mhiche.

**Methodology:** Ambakisye Kuyokwa Mhiche, Ahmed Mohamed Abade.

**Resources:** George Kabona, Ally Hussein.

**Software:** George Kabona.

**Supervision:** Dinah Gasarasi, George Kabona, Upendo John Mwingira, Ahmed Mohamed Abade.

**Validation:** Upendo John Mwingira.

**Visualization:** Ambakisye Kuyokwa Mhiche.

**Writing – original draft:** Ambakisye Kuyokwa Mhiche.

**Writing – review & editing:** Ambakisye Kuyokwa Mhiche, Dinah Gasarasi, Ahmed Mohamed Abade.

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
