## [Decision Letter · Decision Letter 0]

5 Jul 2024

Dear Mr. Mhiche,

Thank you very much for submitting your manuscript "Mass drug administration coverage and determinants of drug uptake for elimination of Onchocerciasis in Ulanga District" for consideration at PLOS Neglected Tropical Diseases. As with all papers reviewed by the journal, your manuscript was reviewed by members of the editorial board and by several independent reviewers. In light of the reviews (below this email), we would like to invite the resubmission of a significantly-revised version that takes into account the reviewers' comments.

We cannot make any decision about publication until we have seen the revised manuscript and your response to the reviewers' comments. Your revised manuscript is also likely to be sent to reviewers for further evaluation.

Sincerely,

Chao Yan

Academic Editor

Jong-Yil Chai

Section Editor

Reviewer's Responses to Questions

**Key Review Criteria Required for Acceptance?**

**Methods**

-Are the objectives of the study clearly articulated with a clear testable hypothesis stated?

-Is the study design appropriate to address the stated objectives?

-Is the population clearly described and appropriate for the hypothesis being tested?

-Is the sample size sufficient to ensure adequate power to address the hypothesis being tested?

-Were correct statistical analysis used to support conclusions?

-Are there concerns about ethical or regulatory requirements being met?

Reviewer #1: -Are the objectives of the study clearly articulated with a clear testable hypothesis stated? YES

-Is the study design appropriate to address the stated objectives? YES

-Is the population clearly described and appropriate for the hypothesis being tested? YES

-Is the sample size sufficient to ensure adequate power to address the hypothesis being tested? YES

-Were correct statistical analysis used to support conclusions? YES

-Are there concerns about ethical or regulatory requirements being met? NO

Reviewer #2: 1) Village Screening and Confidence Intervals: Were all the villages screened during this survey? If not, it may be beneficial to provide 95% confidence intervals to understand the uncertainty around the mean coverage value found in each village, especially as the values are very close to the 85% coverage used as a threshold.

2) Study Area Details: In the first paragraph of the "Study Area" section, it would be beneficial to mention the year when Ov16 prevalence was determined in the district. Also, specify that the 20.7% prevalence mentioned on children is of Ov16.

3) Specific Villages and Rural/Suburban Classification: The study area includes specific villages in Mahenge, specifically Isongo, Uponera, Mawasiliano, and Togo, as indicated in Figure 1. This should be clearly mentioned in the Methods section, not just in the abstract and results. Additionally, it would be beneficial to specify whether these villages are rural or suburban, as rural areas are typically more endemic for onchocerciasis. The methodology mentions that the district Ov16 prevalence stands at 76% in rural areas and 50% in suburban areas, which is relevant information.

4) Study Population Clarification: In the "Study Population" section, it would be useful to clarify what is meant by "participated in the MDA program for at least one (1) year." Does this mean they must have taken ivermectin for at least a year, been eligible for it for at least a year or been approach by a community drug distributer (CDD) for at least a year? Additionally, clarify that the study focuses on the coverage among eligible individuals (excluding pregnant women and children under 5), not the overall population. This distinction is important because WHO recommends 80% coverage of the overall population or coverage of all of the eligible population, especially in historically hyperendemic settings such as Mahenge.

5) Sample Size Estimate: In the "Sample Size Estimate" section, clarify whether the assumption that the proportion of drug uptake during MDA in Mahenge would be 78% refers to the overall population or the eligible population. This needs to be consistent with the study aims.

6) Data Management and Analysis Details: The methodology is mostly comprehensive. The "Data Management and Analysis" section could benefit from further detail. Specify which 95% level was used (e.g., confidence interval method, uncertainty interval, Wilson, Binomial, etc.). Additionally, clarify that "prevalence ratio" refers to the prevalence of taking ivermectin during MDA and that it is an interchangable term with "relative risk". Specify which covariates were included in the Poisson regression (e.g., possible confounders such as age, gender, etc.). While this is partially mentioned in the results, it should be detailed in the methodology.

7) Statistical Methods: The choice of robust statistical methods is commendable. The authors' choice of regression methods, including the robust standard error, could be supported by reasoning either in the methodology or in the discussion as a potential strength and/or limitation.

Reviewer #3: partly, see my general comments

**Results**

-Does the analysis presented match the analysis plan?

-Are the results clearly and completely presented?

-Are the figures (Tables, Images) of sufficient quality for clarity?

Reviewer #1: -Does the analysis presented match the analysis plan? , YES

-Are the results clearly and completely presented? see my general comments

-Are the figures (Tables, Images) of sufficient quality for clarity? Some changes needed

Reviewer #2: 1. Clarification of "±15 years": In "participants was 37.8 ±15 years," please define what "±15 years" represents. It likely means the standard deviation.

2. Correction in Referencing: Instead of referencing "Figure 1," it should be "Table 1."

3. Clarification of WHO Coverage Recommendations: The statement "Drug uptake coverage from all studied villages was below the optimal coverage (85%) recommended by WHO for successful transmission interruption and elimination of Onchocerciasis" needs clarification. Since the coverage in question pertains to the eligible population for taking ivermectin, I ponder if the recommended coverage should be nearly 100% of the eligible population or 80% of the overall population (including those not eligible for ivermectin, such as pregnant women, lactating women during the first week after birth, children under 5 years of age, and seriously ill patients)? If the authors want to keep the 85% threshold, then appropriate references should be mentioned.

4. Define Abbreviations in Tables: Define any abbreviations used in the tables, such as PR (Prevalence Ratio) and MDA (Mass Drug Administration) in Table 1. Also, clarify what "*" signifies, typically indicating statistical significance. Similarly, define CPR, CI (Confidence Interval), and APR (Adjusted Prevalence Ratio) in Table 2.

5. Clarification on Drug Uptake: The statement "In Isongo village, drug uptake was 2.4 times higher in 2017 [PR = 2.41 (2.2-2.3)], while drug uptake in 2018 was 1.2 times higher [PR = 1.18 (1.1-1.3)] compared to that of 2016 respectively" seems inconsistent with the proportions shown in Table 1, which are lower for 2017 and 2018 than for 2016. This discrepancy may relate to the relative risks produced from Poisson Regression (PR), but even if that is true, would it be correct to state that the drug uptake was higher? Please clarify this.

6. Correcting Abbreviation Usage: In "OR = 0.87(0.78-0.96)," should "OR" be "PR"? Please define the abbreviation the first time it is used in the text. Also, define APR.

7. Improving Comparisons: Improve and complete the comparison in the sentence: "Participants aged 15-24 and 25-34 years were 3 times [APR = 2.8 (95% CI: 1.3-6.2)] and 2 times [APR = 2.3 (95% CI: 1.04-4.9)] more likely to take ivermectin, respectively, compared to the adult population aged 45-54 (Table 2)."

8. Consider Merging Categories: In the regression analysis shown in Table 2, consider merging categories for rare values such as "college," "none," and "student." Small sample sizes in these categories might compromise the model output. A rule of thumb is to have at least 10 observations per category per level (yes/no).

9. Clarification of Variables in Table 3: Clarify the variable "Perceived reason for drug uptake" in Table 3 (and correct the table's label from Table 2 to Table 3). What is meant by "prevention effects" and "side effects"? It appears that those who perceived ivermectin as preventive took it less often than those who perceived it as having possible side effects. Also, it is noteworthy how most of those who feared taking ivermectin with alcohol still decided to take the drug. Please provide more context and clarification.

Reviewer #3: partly, see my general comments

**Conclusions**

-Are the conclusions supported by the data presented?

-Are the limitations of analysis clearly described?

-Do the authors discuss how these data can be helpful to advance our understanding of the topic under study?

-Is public health relevance addressed?

Reviewer #1: -Are the conclusions supported by the data presented? YES

-Are the limitations of analysis clearly described? Not sufficient

-Do the authors discuss how these data can be helpful to advance our understanding of the topic under study? Old data is limited their relevance

-Is public health relevance addressed? Limited relevan today because old data

Reviewer #2: 1. Commendation and Clarification on MDA Coverage: I commend the authors on the discussion around MDA coverage and persistent onchocerciasis prevalence and presumable transmission. However, the MDA values seem to be reaching a likely maximum feasible value in the villages. Considering the population usually excluded from receiving ivermectin (pregnant women, lactating women during the first week after birth, children under five, and very sick individuals), the coverage cannot go much higher. Being that Ulanga is one of the most historically endemic settings for onchocerciasis in Tanzania, biannual CDTI is possibly needed to halt transmission. This is indeed a challenge that many settings might face as they approach the elimination goal where the blackfly vector is highly prevalent.

2. Emphasis on Additional Control Interventions: Instead of solely emphasising the need to intensify MDA awareness campaigns targeting less compliant groups in the community to reinforce the benefits of ivermectin in onchocerciasis control and address the community misconceptions about MDA, which might only increase coverage by 5-10% if extremely successful, it might be more effective to advocate for additional control interventions, such as the current biannual CDTI, taking into account the historically high onchocerciasis endemicity of Mahenge and the long history of ivermectin annual delivery.

3. Recent Changes and Comparisons: It was not mentioned that Mahenge recently switched from annual to biannual MDA to further control onchocerciasis. This should be stated. Additionally, referencing recent papers on CDTI coverage in Mahenge for comparison would be very beneficial and expected. An example could be the paper: "Impact of a bi-annual community-directed treatment with ivermectin programme on the incidence of epilepsy in an onchocerciasis-endemic area of Mahenge, Tanzania: A population-based prospective study."

4. Association Between Onchocerciasis and Epilepsy: Given the association between onchocerciasis and epilepsy, known as onchocerciasis-associated epilepsy (OAE), where the onset of epilepsy occurs in children and teenagers aged 3 to 18 years, ivermectin intake has been associated with a decrease in epilepsy incidence in these age groups. This highlights the importance of achieving the highest coverage of ivermectin in the group at risk of developing onchocerciasis-associated epilepsy in Mahenge. This is particularly relevant as many onchocerciasis-endemic sites with high epilepsy, including nodding syndrome, prevalence have the lowest ivermectin coverage in younger age groups. Mahenge was indeed the first place to report the public health problem of nodding syndrome in 1962.

5. Study Limitations: The study limitations should be discussed before the conclusion.

6. Discussion of Other Potential Avenues to Increase Coverage: Being a paper exploring the limitations of ivermectin coverage in the population, it would be valuable to discuss other potential avenues to increase coverage:

6.1) Moxidectin: Another microfilaricidal drug like ivermectin, moxidectin, can be given to children aged 4 years old, complementing ivermectin, which can only be given to children aged five and above. This could help increase coverage among younger children.

6.2) Interventions for Pregnant Women: Exploring potential interventions for pregnant women, who can pass years excluded from ivermectin delivery and represent infection reservoirs, could also be discussed. Potential strategies could include the development of alternative treatments safe for pregnant women or targeted interventions post-pregnancy to ensure they receive ivermectin as soon as it is safe to do so.

Reviewer #3: yes

**Editorial and Data Presentation Modifications?**

Reviewer #1: Old data with limiter relevance

Maybe acceptable if more recent information about the study area is added

Reviewer #2: Missing Reference Numbers: The study appears to have references with numbers up to 72, with many missing numbers in the middle. Additionally, the reference list only goes up to 24. Please ensure that all references are correctly numbered and included in the reference list.

I recommend a thorough review of the entire document for language and punctuation. While the content is generally clear, certain passages could benefit from improved clarity and precision. For instance, here are some abstract corrections:

1.1. Correct "Onchocerciasis remain to be an important" to "Onchocerciasis remains an important."

1.2. In the sentence "However, current reports indicate high prevalence of Onchocerciasis in both human and vector species probably because of poor treatment coverage indicating limited evidence for transmission," two points need clarification:

- It seems that poor treatment coverage is being identified as the cause of limited evidence for active transmission.

- The phrase "limited evidence for transmission" likely means the opposite of what is intended. It seems the authors want to express that there is preliminary evidence of persistent onchocerciasis transmission, which likely sustains the observed high prevalence.

1.3. Add a comma after "Morogoro" in "Ulanga District, Morogoro, Tanzania from April-June 2019." Remove the space in "25.5 %" and add a comma after "Mawasiliano (68%)."

1.4. In "Low coverage below the WHO optimal recommended coverage," the words "low" and "below" are redundant. I suggest removing "low."

Reviewer #3: major revision

**Summary and General Comments**

Reviewer #1: The papers addresses an important challenge for the elimination of onchocerciasis in Tanzania: to obtain a sufficient high uptake of ivermectin.

The paper is well written and the results are interesting. However the main problem with the paper is that the study was done in 2019. This limits the relevance of the data for operational use because we need to know what the ivermectin coverage is today, in 2024. In the mean time in the study area CDTI has been switched from annual to bi-annual distributions. This certainly had an effect on the current uptake of ivermectin in the area. It also had an effect on the incidence of epilepsy in the area.

Important to mention this in the discussion. See ref

Bhwana D, Amaral LJ, Mhina A, Hayuma PM, Francis F, Siewe Fodjo JN, Mmbando BP, Colebunders R. Impact of a bi-annual community-directed treatment with ivermectin programme on the incidence of epilepsy in an onchocerciasis-endemic area of Mahenge, Tanzania: A population-based prospective study. PLoS Negl Trop Dis. 2023 Jun 28;17(6):e0011178.

I propose to explain better why it was important in 2019 to investigate ivermectin coverage in the Ulanga District in Tanzania and not in other onchocerciasis endemic areas in Tanzania.

The authors report the 2009 and 2017 survey data in Ulanga. However it is important to specify that some of the data were obtained in selected rural villages, different from the villages investigated for ivermectin coverage in 2019.

On the map Figure 1 it would be interesting to indicate the localisation of the 2019 study villages together with those investigated in the epilepsy prevalence surveys in 2017-18.

A limitation of the study is that children < 15 years of age were not included. This is important because they are at risk for developing onchocerciasis-associated epilepsy and therefore it is important that a high ivermectin coverage is reached in the age group.

Methods

Little is said about the questionnaire and how it was used. How were the house visits done? What happened if in a selected household nobody was home? Secondary data collection was done in 2 health facilities. Maybe I missed it but I do not see the results clearly explained in the text. Were the data that were obtained different from those obtained by interviewing the study participants?

Minor comments

multi variate analysis should this be multivariable analysis?

Table 1 Togo should be written in bold

All Tables: Harmonise spaces between numbers and percentages

Table 2. Onchocerciasis Targeted health education: not clear what the following variables stand for:

Hygiene and Ocho ? Other but not Ocho?

I propose to add the questionnaire as supplementary material to better understand these variables.

Reviewer #2: The study by Mhiche et al. addresses a crucial public health issue by examining mass drug administration (MDA) coverage and the factors influencing drug uptake in onchocerciasis-endemic regions. The research demonstrates a thorough understanding of onchocerciasis and its elimination guidelines, providing valuable insights into the challenges and successes of MDA programs.

While the study is comprehensive and informative, several areas require improvement to enhance clarity and accuracy. These include detailed explanations of the methodology, addressing discrepancies in the results, incorporating recent changes in MDA strategies, and ensuring all references are correctly numbered and included.

With these revisions, the manuscript could make a significant contribution to onchocerciasis research. I acknowledge the authors' hard work and dedication to improving public health outcomes.

1) Abstract remakrs:

1.1) Clarification of "±15 years": Please define what "±15 years" represents in the result section of the abstract. Is this a standard deviation, or another statistical measure?

1.2) WHO Optimal Coverage: The statement "below the optimal coverage recommended by WHO (85%) for successful transmission interruption" needs clarification. From my understanding, the WHO guideline for elimination is to deliver ivermectin to at least 80% of the population. However, in the Americas, the guideline specifies at least 85% of the eligible population. If you are referring to the latter, please clarify that you are working with the eligible population and not the overall population. However, keep in mind that recommendations for the Americas might not be applicable to sub-Saharan Africa, as the latter have more competent vectors and, therefore, higher endemicities.

1.3) Terminology in Association: In the phrase "decreased drug uptake [APR = 12(95% CI: 2.4-60.9), p<0.05)] was attributable," the word "attributable" implies causation, which cannot be formally assessed with a Poisson regression. I suggest using terms like "associated with" or "linked to" instead.

1.4) Correct "Adjusted Prevalence Ration" to "Adjusted Prevalence Ratio". Also, consider that, the terminology "adjusted relative risk" is more often used.

2) Introduction Remarks:

2.1) Clarification of Geographic Focus: The phrase "and in Tanzania in particular" should be corrected to "including Tanzania" to avoid implying that onchocerciasis in Africa is particularly prevalent in Tanzania.

2.2) Detailed Explanation of Epilepsy Co-Endemicity: In the last paragraph of the introduction, epilepsy is briefly mentioned as co-endemic with parasitic infections such as onchocerciasis. A more detailed explanation and referencing would be beneficial. Specifically, Mahenge is one of the most well-known foci demonstrating the strong association between onchocerciasis and epilepsy, including nodding syndrome. In Tanzania, Mahenge is clearly the most well-known focus for nodding syndrome. This context can support the authors' argument that the high prevalence and incidence of epilepsy in the area may further indicate active transmission of onchocerciasis. It has been shown in several endemic settings, including Mahenge, that the implementation and strengthening of onchocerciasis control efforts are followed by a significant decrease in the incidence of epilepsy, including nodding syndrome. Providing this detailed explanation and referencing relevant studies will strengthen the introduction and support the study's rationale.

2.3) Historic Prevalence Information: Including specific information, with numbers if available, on historic prevalence of Mahenge would provide valuable context for understanding the current study findings. It would help readers appreciate the progress made and the challenges that remain in controlling onchocerciasis.

Reviewer #3: Comments

1. Under abstract section how the determinants were explored, better to use other appropriate term

2. Correct some typos throughout the paper

3. Why the study was this much late, are there no changes since that (2019).

4. Eligibility criteria is not clear, what was sampling and study unit, how many from each household?,

5. Regarding sample what does 78% as previously determined mean?

6. How was 2 data collection methods (primary and secondary) used together?

7. Did you included qualitative data, “in addition, an interview guide was developed to guide the key informant interviews.”, if so how, for what purpose, who were the participants….?

8. Why was this analysis selected, ‘Following descriptive analysis, generalized linear model with modified Poisson regression was used to estimate prevalence ratio.’ And did the assumptions were checked and if so give us supplementary files

9. What is your base for your categorization for variables like age, occupation, etc.

10. Title needs revision to make concise, what if Mass drug administration coverage and its determinants….think critically over it

11. Remove repetition from the abstract, remove interpretation from result

12. How you see this finding with very wide confidence interval like (CI:2.9-60.9)….

13. Why Modified Poisson regression was chosen compared to others?

14. Better to put sampling procedure in figure to make it clear for readers by showing the stages

15. Did qualitative method included? How? The purpose, if so how the data was analyzed and interpreted? There is a statement w/c talks about KII.

16. Measurement and operational definitions should be presented clearly?

17. Discussion lacks depth, there are also incomplete sentences, …need major revision

18. Clearly justify way to control bias, better to include strobe checklist

PLOS authors have the option to publish the peer review history of their article (<a href="https://journals.plos.org/plosntds/s/editorial-and-peer-review-process#loc-peer-review-history" target="_blank">what does this mean?). If published, this will include your full peer review and any attached files.

**Do you want your identity to be public for this peer review?** For information about this choice, including consent withdrawal, please see our Privacy Policy .

Reviewer #1: Yes: Colebunders Robert

Reviewer #2: Yes: Luís-Jorge Amaral

Reviewer #3: No
---

## [Decision Letter · Decision Letter 1]

26 Feb 2025

Dear Dr. Mhiche,

Response to Reviewers
Revised Manuscript with Track Changes
Manuscript

Shaden Kamhawi

co-Editor-in-Chief

Paul Brindley

co-Editor-in-Chief

**Journal Requirements:**

1) Please provide an Author Summary. This should appear in your manuscript between the Abstract (if applicable) and the Introduction, and should be 150-200 words long. The aim should be to make your findings accessible to a wide audience that includes both scientists and non-scientists. Sample summaries can be found on our website under Submission Guidelines:

2) We have noticed that you have uploaded Supporting Information files, but you have not included a list of legends. Please add a full list of legends for your Supporting Information files after the references list.

**Reviewers' comments:**

**Key Review Criteria Required for Acceptance?**

**Methods** :

-Are the objectives of the study clearly articulated with a clear testable hypothesis stated?

-Is the study design appropriate to address the stated objectives?

-Is the population clearly described and appropriate for the hypothesis being tested?

-Is the sample size sufficient to ensure adequate power to address the hypothesis being tested?

-Were correct statistical analysis used to support conclusions?

-Are there concerns about ethical or regulatory requirements being met?

Reviewer #1: (No Response)

Reviewer #2: 1) Please mention the reference where this important information was obtained: "The district OV16 prevalence as of

2017 stands at 76% and 50% in rural and suburban areas respectively."

2) Remove either in " These villages were classified as either rural, where onchocerciasis prevalence tends to be higher (Figure 1)"

Reviewer #3: (No Response)

**Results** :

-Does the analysis presented match the analysis plan?

-Are the results clearly and completely presented?

-Are the figures (Tables, Images) of sufficient quality for clarity?

Reviewer #1: (No Response)

Reviewer #2: 1) Table 1, How are the numbers of people surveyed in each village higher than the study sampled population of 502 participants? Please clarify this in the methodology/results where this data comes from reference it if applicable.

2) should be "crude" in "CPR= Cruse Prevalence Ratio"

Reviewer #3: (No Response)

**Conclusions** :

-Are the conclusions supported by the data presented?

-Are the limitations of analysis clearly described?

-Do the authors discuss how these data can be helpful to advance our understanding of the topic under study?

-Is public health relevance addressed?

Reviewer #1: (No Response)

Reviewer #2: It should be mentioned in the study that Mahenge is currently under biannual MDA (ref 16), which this study results support.

I am confused between staying that two of the 4 villages met the 80% WHO recommended coverage for onchocerciasis elimination, but then, in table 1, all of them have <<80% coverage on the population data. As stated above, I am also still confused between the data collected from the 502 participants (2019) and the one collected from a larger population (Table 1, 2016-2018). This should be clarified. Does the study seem to suggest that coverage increased between 2016-2018 and 2019?

The coverages of table 1 (2016-2018) and the participants in 2019, being from the same villages and assuming independence, should be statistically analysed in the results and interpreted in the discussions, otherwise, table 1 seems to be of little to no relevance for the discussion of this manuscript.

Reviewer #3: (No Response)

**Editorial and Data Presentation Modifications?**

Reviewer #1: (No Response)

Reviewer #2: Title: I would consider specifying in the title that the study was conducted in Tanzania (e.g., "Mass drug administration coverage and its determinants for elimination of Onchocerciasis in Ulanga District, Tanzania).

Abstract:

1) In the abstract, is there a reason for describing age with the mode instead of the mean/median?

2) The results section of the abstract do not read very well and would benefit from revision, especially ""MDA coverage for studied villages was Mawasiliano (68%, CI: 59.3 – 75.6), Uponera (83%, CI; 76.6 – 89.6), Isongo (84 .8 %,CI: 77.3 – 90.1) and Togo (79%, CI: 70.1 – 85.8)

3) correct punctuation in " and newcomers to the village. emphasize the need to intensify"

4) In the abstract, I could be beneficial information to mention that Ulanga is hyperendemic (or very endemic) for onchocerciasis prior to control, which means that indeed MDA needs to be particularly of high coverage.

Introduction:

1) Conduct a revision for punctuation (e.g., remove commas and doubl3 spaces in "Onchocerciasis (river blindness) is a vector-borne parasitic disease caused by , Onchocerca volvulus, and...; remove unnecessary capitalisation in Onchocerciasis Control"; add "a" to "rea" in "filaria worms in the rea, suggesting").

2) In "attain and maintain the recommended MDA treatment coverage and demonstrate the interruption of

transmission among vector species (6)." mention the recommended MDA treatment coverage (80% by WHO) and specify it is the coverage of the overall population.

3) Please clarify the last part of this important sentence: ". A baseline assessment onchocerciasis conducted in 1997 indicated nodule prevalence of 45% and 95% Ulanga districts(14). "

4) Mention the reference used to write this sentence please "Onchocerciasis nodule prevalence was 2.3%, while the positivity rate for OV16 rapid test was 76.5% and the risk of transmission among children aged 6-9 years was 20.7%." Also, clarify what is meant by "risk", which might be misleading, if it is seroprevalence among children 6-9, then it should be stated so. Also, please specify the year of the study. Then, if the authors wish, they can state that this high seroprevalence in young children, taking into account that onchocerciasis is a chronic infection that can last over 10 years, suggests that there is still high O. volvulus transmission at that time.

5) I would suggest specifying these numbers: "On the other side, literature shows the black fly carries infective parasites and the infection rate is clearly above the threshold for transmission interruption within Ulanga district (23). "

Reviewer #3: (No Response)

**Summary and General Comments** :

Reviewer #1: I checked the responses of the authors to my comments.

For me the revised paper can be published.

Reviewer #2: I congratulate the authors in addressing the reviewers' comments and adding valuable background information to their study, especially regarding onchocerciasis epidemiology in Mahenge. I believe the manuscript is in better shape to be accepted, pending minor revisions of scientific content and punctuation (e.g., double spaces, unnecessary capitalisations, missing words).

Reviewer #3: To improve validity you mentioned we used standardized tool for both survey and qualitative, how this was ensured? Which types of validity you ensured for survey also better to tell us how reliability was ensured, similarly for qualitative guide?

PLOS authors have the option to publish the peer review history of their article (what does this mean? ). If published, this will include your full peer review and any attached files.

**Do you want your identity to be public for this peer review?** For information about this choice, including consent withdrawal, please see our Privacy Policy .

Reviewer #1: **Yes: ** Robert Colebunders

Reviewer #2: **Yes: ** Luís-Jorge Amaral

Reviewer #3: **Yes: ** Daba Abdissa

**Figure resubmission:****Reproducibility:** To enhance the reproducibility of your results, we recommend that authors of applicable studies deposit laboratory protocols in protocols.io, where a protocol can be assigned its own identifier (DOI) such that it can be cited independently in the future. Additionally, PLOS ONE offers an option to publish peer-reviewed clinical study protocols. Read more information on sharing protocols at https://plos.org/protocols?utm_medium=editorial-email&utm_source=authorletters&utm_campaign=protocols

---

## [Editor Report · Decision Letter 2]

27 Mar 2025

Dear Mr. Mhiche,

We are pleased to inform you that your manuscript 'Mass Drug Administration Coverage and Its Determinants for the Elimination of Onchocerciasis in Ulanga District, Tanzania' has been provisionally accepted for publication in PLOS Neglected Tropical Diseases.

Best regards,

Chao Yan

Academic Editor

Jong-Yil Chai

Section Editor

Shaden Kamhawi

co-Editor-in-Chief

Paul Brindley

co-Editor-in-Chief

---

## [Editor Report · Acceptance letter]

Dear Mr. Mhiche,

We are delighted to inform you that your manuscript, "Mass Drug Administration Coverage and Its Determinants for the Elimination of Onchocerciasis in Ulanga District, Tanzania," has been formally accepted for publication in PLOS Neglected Tropical Diseases.

Best regards,

Shaden Kamhawi

co-Editor-in-Chief

Paul Brindley

co-Editor-in-Chief
